# Association between Diabetic Peripheral Neuropathy as Measured Using a Point-of-Care Sural Nerve Conduction Device and Urinary Albumin Excretion in Patients with Type 2 Diabetes

**DOI:** 10.3390/jcm12124089

**Published:** 2023-06-16

**Authors:** Tatsuya Fukuda, Akiko Fujii, Taro Akihisa, Naoya Otsubo, Masanori Murakami, Tetsuya Yamada, Chisato Maki

**Affiliations:** 1Department of Endocrinology and Metabolism, Tokyo Metropolitan Okubo Hospital, Tokyo 160-8488, Japan; 2Department of Molecular Endocrinology and Metabolism, Graduate School of Medical and Dental Sciences, Tokyo Medical and Dental University, Tokyo 113-8510, Japan

**Keywords:** DPN-Check, urinary albumin excretion, type 2 diabetes, diabetic peripheral neuropathy, simplified diagnostic criteria

## Abstract

Background: It is not well known whether diabetic peripheral neuropathy diagnosed using a non-invasive point-of-care nerve conduction device called DPN-Check^®^ is associated with diabetic nephropathy. Thus, we aimed to evaluate the association of diabetic peripheral neuropathy with urinary albumin excretion in patients with type 2 diabetes using DPN-Check^®^. Methods: This retrospective observational study included 323 Japanese patients with type 2 diabetes. The urinary albumin-to-creatinine ratio in a spot urine sample was defined as urinary albumin excretion. Multiple linear regression analysis was used to determine the association of DPN-Check^®^-determined diabetic peripheral neuropathy with urinary albumin excretion. Results: Patients with DPN-Check^®^-determined diabetic peripheral neuropathy had significantly higher urinary albumin excretion than those without, while there was no difference in urinary albumin excretion between patients with and without diabetic peripheral neuropathy determined by simplified diagnostic criteria. In the multivariate model, the DPN-Check^®^ determined that diabetic peripheral neuropathy was significantly associated with urinary albumin excretion even after adjustment for covariates (standardized β, 0.123; *p* = 0.012). Conclusions: Our study found a significant association between diabetic peripheral neuropathy diagnosed using DPN-Check^®^ and urinary albumin excretion in patients with type 2 diabetes.

## 1. Introduction

Diabetic peripheral neuropathy (DPN) is a common and debilitating complication of diabetes mellitus, affecting up to 50% of patients with diabetes [1,2,3]. It is characterized by nerve damage, leading to a variety of symptoms such as pain, numbness, and loss of sensation in the extremities, particularly in the lower limbs [4,5,6]. Diabetic peripheral neuropathy is frequently regarded as one of the earliest and most common microvascular complications in individuals with diabetes [4]. Therefore, early detection and effective management of DPN are crucial to prevent further complications and improve quality of life for diabetic patients [7].

Urinary albumin excretion (UAE) serves as an early diagnostic marker for kidney damage, which is another prevalent complication in diabetes [8,9,10]. There has been growing evidence suggesting a potential association between diabetic neuropathy and UAE, highlighting the interconnected nature of these complications [11,12]. Accurately identifying and characterizing DPN may not only help to manage this debilitating condition more effectively, but also may be beneficial in identifying patients at risk for future declines in kidney function. A comprehensive understanding of the relationship between DPN and UAE could lead to improved diagnostic and therapeutic approaches, ultimately reducing the burden of microvascular complications in diabetic patients.

One of the well-established methods to assess DPN is nerve conduction velocity (NCV) testing. NCV testing evaluates the functionality of peripheral nerves by measuring the speed at which electrical signals travel along the nerves [13]. However, traditional NCV testing has several limitations, such as being time-consuming, expensive, and requiring skilled personnel to perform the test [14]. These factors can limit the accessibility and widespread use of NCV testing in clinical settings, especially in resource-limited areas. To address this issue, there is a need for a simpler, more accessible diagnostic tool to evaluate DPN in a clinical setting. Recently, a non-invasive point-of-care nerve conduction device called DPN-Check^®^ (Neurometrix Inc., Waltham, MA, USA) has been introduced. This device identifies impairments in sural nerve conduction, such as reduced amplitude (AMP) and conduction velocity (CV), during a sural nerve conduction study [15,16]. Perkins BA reported that the sensitivity and specificity of normal and abnormal sural nerve AMP measured by DPN-Check^®^ for the detection of diabetic sensorimotor polyneuropathy, as defined by standard clinical and electrophysiological criteria, were 92% and 82%, respectively [15]. In addition, Shibata Y reported that, in Japanese patients with diabetes, significant correlations were observed in the CV (r = 0.7734) and AMP (r = 0.6155) obtained by NCS testing using a standard electromyography system and DPN-Check^®^ [17]. Given this, DPN-Check^®^ has the potential to be a more convenient and accessible tool for DPN diagnosis, especially in primary care settings. Despite the promise of DPN-Check^®^ as an alternative diagnostic tool for DPN, there are no reports examining the association between DPN diagnosed by the DPN-Check^®^ and urinary albumin excretion. Urinary albumin excretion is a clinically relevant marker, as it is associated with the development and progression of diabetic nephropathy, another common complication of diabetes. Investigating the relationship between DPN diagnosed by DPN-Check^®^ and urinary albumin excretion could provide valuable insights into the clinical utility of this novel diagnostic device.

Thus, in this study, we aimed to evaluate the association of DPN, determined using DPN-Check^®^, with UAE in patients with type 2 diabetes.

## 2. Materials and Methods

### 2.1. Study Design and Population

This was a single-center retrospective observational study conducted to assess the association of DPN determined using DPN-Check^®^ with UAE in Japanese patients with type 2 diabetes. All investigations used data obtained from hospital records. Patients aged ≥20 years who had visited the Tokyo Metropolitan Okubo Hospital between 1 May 2015 and 31 March 2023 were included. Patients were eligible if they were diagnosed with type 2 diabetes according to the criteria of the Japan Diabetes Society (JDS) [18]. Regarding the enrolled patients, no upper age limit was established. Exclusion criteria included severe renal impairment (estimated glomerular filtration rate (eGFR) < 15 mL/min/1.73 m^2^, or undergoing renal replacement therapy), pregnancy, and diseases other than diabetes that could cause neuropathy (stroke or spinal canal stenosis). As a result, this study included 323 patients, ranging in age from 31 to 96 years. 

The present study was carried out in accordance with the Declaration of Helsinki. The protocol for this multicenter collaborative research was collectively reviewed by the ethical review committee of the Tokyo Metropolitan Health and Hospitals Corporation, Okubo Hospital, and was approved (2020–2024). Written informed consent was not obtained due to the retrospective nature of the study. We provided patients with the opportunity to opt out by displaying an outline of the analysis on the hospital’s website.

### 2.2. Clinical Data Collection

We collected data on comorbidities (diabetic retinopathy (DR) and cardiovascular disease (CVD)), demographics (age, sex, body mass index, duration of diabetes, and smoking status), laboratory test results, and medication from the electronic medical records. DR was classified into three groups: patients without DR (NDR), patients with simple DR (SDR), and patients with proliferative DR (PDR), according to the medical records. 

CVD is defined as the history of unstable angina, myocardial infarction, percutaneous coronary intervention, coronary bypass grafting, angioplasty, or major amputation due to peripheral arterial disease). The ankle brachial pressure index (ABI) and brachial-ankle pulse wave velocity (baPWV) were measured by trained technicians using an blood pressure pulse wave testing device (FORM-5, Fukuda Colin Co., Ltd., Tokyo, Japan) as markers for atherosclerosis. The average values of baPWV from both the right and left side were used for analyses. The coefficient of variation of RR intervals (CVR-R) in the resting state was measured by technicians to assess autonomic nervous function using ECG-2320 (Nihon Kohden Co., Ltd., Tokyo, Japan). Systolic and diastolic blood pressure (SBP and DBP) was measured while the subjects were seated, after rest period of at least 2 min prior to the DPN-Check^®^ assessment.

The baseline clinical data were obtained on the closest day within one year of the date when neuropathy was determined using DPN-Check^®^. As laboratory test results, Hemoglobin A1c (HbA1c), triglycerides, high density lipoprotein (HDL) cholesterol, low density lipoprotein (LDL) cholesterol, aspartate transaminase (AST), alanine transaminase (ALT), γ-glutamyl transpeptidase (γ-GTP), uric acid (UA), urinary albumin-to-creatinine ratio (ACR) and estimated glomerular filtration rate (eGFR) were collected from the patients’ medical records. HbA1c was measured using the latex agglutination method. Serum creatinine was measured using an enzymatic method, and GFR was estimated according to CKD-EPI equation modified by a Japanese coefficient [19]. GFR categories were defined according to the KDIGO criteria as follows: G1, eGFR ≥ 90; G2, eGFR 60–89; G3a, eGFR 45–59; G3b, eGFR 30–44; G4, eGFR 15–29 [20]. Urinary albumin and creatinine concentrations were measured using turbidimetric immunoassays and the enzymatic method, and ACR (mg/g) was calculated for the assessment of albuminuria in a spot urine sample. Normoalbuminuria was defined as ACR < 30 mg/gCr, microalbuminuria was defined as ACR 30–299 mg/gCr, and macroalbuminuria was defined as ACR ≥ 300 mg/gCr.

### 2.3. Definition of Diabetic Peripheral Neuropathy

Sural nerve conduction velocities were measured using DPN-Check^®^ HDN-1000 (Neurometrix Inc., Waltham, MA, USA), which was consistently performed by one nurse (A.F.) specializing in diabetes care, under the direction and supervision of diabetologists certified by JDS, throughout the study period. The diabetologists had completed a training course by the manufacturer of DPN-Check^®^; however, they had not completed a course in electromyography. The averaged values of both amplitude (μV, AMP) and conduction velocity (m/s, CV) were calculated from bilateral results. Undetectable nerve conduction was considered AP 0 μV and CV 0 m/s for DPN diagnosis. DPN-Check^®^ identified DPN if either CV or AMP was below the threshold. The threshold values were determined based on the previous study [21] as follows: AMP threshold = 12.62 − 0.103 × age (years), CV threshold = 94.88 − 0.148 × age (years) − 0.231 × height (cm). We defined myelopathy and axonopathy as neuropathies showing abnormalities of CV and AMP, respectively. Severe DPN-Check^®^ was defined as if AMP < 2 μV, according to Baba’s classification [22].

In addition, as another way to determine diabetic peripheral neuropathy, simplified diagnostic criteria defined DPN based on “simplified diagnostic criteria of diabetic polyneuropathy and suggested staging”, as proposed by the consensus of the Japanese study group of diabetic neuropathy [22]. To assess the simplified diagnostic criteria determining DPN, at least two of the following three criteria had to be met: subjective symptoms (numbness, pain, or dysesthesia in bilateral lower extremities); diminished bilateral Achilles tendon reflexes, including decreased and absent ATR; or diminished bilateral vibratory sensation at the malleolus medialis (<10 s when using a tuning fork at 128 Hz). Sural nerve conduction velocities were measured using DPN-Check^®^ by trained technicians. These three aforementioned examinations were consistently performed by one nurse (A.F.) at the same time as the DPN-Check^®^ measurement.

### 2.4. Statistical Analysis

Statistical analysis was carried out using SPSS (version 21.0; IBM Corp., Armonk, NY, USA). Data are presented as mean ± standard deviation, median with interquartile range, or percentage according to data distribution. Categorical variables were compared using the chi-squared test or Fisher’s exact test. The differences were analyzed using the *t*-test or Mann–Whitney U test, as was appropriate for the data distribution. Multivariate linear regression analysis was used to determine the associations between the variables, including DPN-Check^®^-determined DPN, and log_10_-transformed ACR. In multiple linear regression analysis, the selection of the variables to incorporate into the model was performed using a stepwise procedure. The following variables were incorporated as covariates in the multivariate linear regression: age, male sex, body mass index, current smoker, duration of diabetes, CVD, SDR, PDR, baseline medication, laboratory parameters, and physiological testing data. *p* < 0.05 was regarded as statistically significant.

## 3. Results

The patient characteristics, according to DPN status as determined by DPN-Check^®^, are shown in Table 1. Of the 323 patients, 158 (48.9%) had DPN-Check^®^-determined DPN, and 158 (48.9%) had simplified diagnostic criteria-determined DPN. Of the 323 patients, there were 181 patients (56.0%) with normoalbuminuria, 99 patients (30.7%) with microalbuminuria, and 43 patients (13.3%) with macroalbuminuria. Patients with DPN-Check^®^-determined DPN had a higher BMI and a higher percentage of patients with CVD, SDR, and PDR than those without. With regard to the medications patients were receiving, patients with DPN-Check^®^-determined DPN were more likely than those without to be using insulin, beta blockers, and anti-platelet agents. As for laboratory parameters, patients with DPN-Check^®^-determined DPN tended to have lower HDL cholesterol and higher ACR levels than those without. The prevalence of patients classified as G3b was significantly higher in the patients with DPN-Check^®^-determined DPN. Patients with DPN-Check^®^-determined DPN had lower values of CV and AMP than those without. Similarly, patients with DPN-Check^®^-determined DPN had lower CVR-R values than those without. The percentage of patients with diminished vibratory sensation, which is one of the components of simplified diagnostic criteria for DPN, was significantly higher in the patients without DPN-Check^®^-determined DPN. The proportion of patients with macroalbuminuria was significantly higher in patients with DPN-Check^®^-determined DPN than in those without DPN-Check^®^-determined DPN (20% vs. 7%, *p* < 0.001). There were no differences in ACR between patients with severe DPN-Check^®^-determined DPN (39.4 mg/gCr [11.0–174.0]) and those with non-severe DPN-Check^®^-determined DPN (23.0 mg/gCr [11.0–94.0]).

Figure 1A shows the association between DPN and ACR as determined using each of the DPN-Check^®^ and simplified diagnostic criteria, respectively. Patients with DPN-Check^®^-determined DPN had significantly higher ACR levels than those without (*p* < 0.001), while there were no differences in ACR between patients with and without simplified diagnostic criteria-determined DPN. Figure 1B shows log_10_-transformed ACR values in patients without neuropathy (n = 165, 1.2 [0.8–1.7]), with myelopathy only (n = 25, 1.4 [1.0–2.5]), with axonopathy only (n = 37, 1.4 [1.0–2.0]), and with both myelopathy and axonopathy (n = 96, 1.6 [0.1–2.3]), determined by the values of AV and AMP in DPN-Check^®^. Statistically significant differences between the four groups were detected by ANOVA (*p* < 0.001). The post-hoc test showed that log_10_-transformed ACR values were significantly higher in patients with myelopathy only and in those with both myelopathy and axonopathy than that in patients without neuropathy.

Figure 2 shows the Spearman’s rank correlation coefficients of CV (Figure 2A) and AMP (Figure 2B), obtained by DPN-Check^®^, for the left and right sural nerve, and of CV and AMP with log10-transformed ACR (Log_10_-ACR). The CV and AMP of the left and right sural nerves each showed a significant correlation (CV, Spearman’ Rho = 0.504, *p* < 0.001; AMP, Spearman’ Rho = 0.606, *p* < 0.001). As shown in Figure 2C,D, both CV and AMP were negatively correlated with Log_10_-ACR (CV, Spearman’ Rho = −0.286, *p* < 0.001; AMP, Spearman’ Rho = −0.235, *p* < 0.001). In the analyses of patients with DPN-Check^®^-determined DPN, CV and AMP were not significantly associated with Log10-transformed ACR (CV, Spearman’ Rho = −0.084, *p* = 0.294; AMP, Spearman’ Rho = −0.070, *p* = 0.349).

Figure 3 shows Spearman’s rank correlation coefficients of conduction velocity and amplitude in DPN-Check^®^, along with other clinical parameters. CV was negatively associated with age (Spearman’ Rho = −0.206, *p* < 0.001), HbA1c (Spearman’ Rho = −0.117, *p* = 0.036), HDL (Spearman’ Rho = 0.144, *p* = 0.009), ALT (Spearman’ Rho = 0.175, *p* = 0.002), eGFR (Spearman’ Rho = 0.214, *p* < 0.001), Log_10_-transformed ACR (Spearman’ Rho = −0.286, *p* = < 0.001), CVR-R (Spearman’ Rho = 0.136, *p* = 0.014), and the duration of diabetes (Spearman’ Rho = −0.261, *p* < 0.001).

AMP was negatively associated with age (Spearman’ Rho = −0.273, *p* < 0.001), eGFR (Spearman’ Rho = 0.227, *p* < 0.001), Log_10_-transformed ACR (Spearman’ Rho = −0.230, *p* < 0.001), baPWV (Spearman’ Rho = −0.178, *p* = 0.001), the duration of diabetes (Spearman’ Rho = −0.196, *p* < 0.001), and CV (Spearman’ Rho = 0.687, *p* < 0.001).

Table 2 shows the multiple regression analysis models examining the association between DPN-Check^®^-determined DPN and urinary albumin excretion. In univariate models, DPN-Check^®^-determined DPN was significantly associated with ACR, and this was consistent across analyses by gender (standardized β, 0.235, *p* < 0.001 for total patients; standardized β, 0.215, *p* = 0.001 for male; standardized β, 0.299, *p* = 0.004 for female). In the total patient group, DPN-Check^®^ determined that DPN and ACR were significantly associated in the age- and gender-adjusted model, even after adjusting for age and gender (standardized β, 0.235; *p* < 0.001). This result was consistent across analyses by gender (gender analyses were adjusted for age only (standardized β, 0.215, *p* = 0.001 for male; standardized β, 0.332, *p* = 0.005 for female). In the multivariate model, the variables significantly associated with ACR were selected using a stepwise method, and even after adjusting for this, the DPN-Check^®^ determined that DPN and ACR were significantly associated (standardized β, 0.123; *p* = 0.012). In the gender analysis, the association between the DPN-Check^®^-determined DPN and ACR was attenuated in males (standardized β, 0.085; *p* = 0.145), while the association between determined DPN and ACR remained significant in women (standardized β, 0.214; *p* = 0.010). In the multivariate model, eGFR was negatively associated with ACR in both total patients and male patents. TG, PDR, SBP, the usage of statins, insulin, and GLP1-Ras were associated with ACR in both total patients and male patents.

On the other hand, the usage of UA-lowering agents and beta blockers, CVR-R, and the duration of diabetes were associated with ACR in female patients. DPN determined by the simplified diagnostic criteria was not consistently selected by the stepwise procedure in the total patient group, male patients, or female patients.

Table 3 shows the multivariate regression analysis models examining the association between abnormalities in CV, AMP, and ACR. In the univariate model, abnormal CV was significantly correlated with ACR (standardized β, 0.162; *p* = 0.018), although this was not the case for abnormal AMP (standardized β, 0.101; *p* = 0.139). Abnormal CV was significantly associated with ACR, even after adjustment for age and gender (standardized β, 0.150; *p* = 0.029). Abnormal CV was consistently associated with ACR after further adjustment for covariates selected by the stepwise procedure in the multivariate model (standardized β, 0.128; *p* = 0.022).

## 4. Discussion

In this study, we investigated the association between DPN diagnosed using DPN-Check^®^, a non-invasive point-of-care nerve conduction device, and UAE in patients with type 2 diabetes. Our results demonstrated that patients with DPN-Check^®^-determined DPN had significantly higher ACR than those without, suggesting a possible link between diabetic peripheral neuropathy and kidney damage in patients with type 2 diabetes. This association remained significant in female patients after adjusting for confounding factors, while the association was attenuated in male patients. Our results may indicate that using DPN-Check^®^ for diabetic peripheral neuropathy is useful for identifying patients at higher risk of renal function decline, more so than simplified diagnostic criteria based on symptoms and physical examination.

The significant association between DPN-Check^®^-determined DPN and ACR which we observed in our study is consistent with previous research that has reported an association between diabetic peripheral neuropathy and UAE [11,12]. As for the mechanisms underlying diabetic peripheral neuropathy and nephropathy, hyperglycemia, oxidative stress, and inflammation have been reported to be contribute to their development and progression [1,23,24]. Hyperglycemia, a major risk factor for both conditions, can lead to the formation of advanced glycation end products (AGEs) [25]. AGEs have been implicated in the pathogenesis of both diabetic peripheral neuropathy and nephropathy, as they can induce oxidative stress and inflammation, exacerbating the damage to nerves and kidneys [13,26]. Since we did not measure AGEs in our study, and there are few clinical reports that investigate the relationship between diabetic peripheral neuropathy diagnosed using DPN-Check^®^ and ACR in which the concentration of AGEs is involved, further studies are warranted to determine whether AGEs are elevated in patients with UAE and diabetic peripheral neuropathy.

Clinical assessment of diabetic neuropathy typically involves evaluating a patient’s self-reported symptoms and conducting a physical examination [27]. Physical examination findings, including the Achilles tendon reflex, can provide information about the integrity of the sensory and motor nerves; however, clinical assessments can be subjective, and their sensitivity and specificity for detecting diabetic neuropathy may vary depending on the examiner’s experience and the severity of the condition [14,28]. On the other hand, NCV studies provide a more objective and quantitative assessment of nerve function in diabetic neuropathy, and NCV studies can detect abnormalities in nerve function even in the early stages of diabetic neuropathy when clinical symptoms and signs may be absent or subtle [29,30,31]. In our study, DPN determined by simplified diagnostic criteria was not associated with ACR, whereas DPN-Check^®^-determined DPN was independently associated with UAE. Considering this, the NCV study may be more useful than determining DPN via simplified diagnostic criteria, which are based on patient’s subjective symptoms. Decreased Achilles tendon reflex and vibration sensation are involved in the diagnosis of diabetic neuropathy in terms of identifying patients at high risk of renal function decline, and DPN-Check^®^ may be a very useful tool for easily studying NCV.

In our study, CV abnormality was more robustly associated with albuminuria excretion than AMP abnormality, as demonstrated in Table 3. The AMP and CV of the sural nerve are quantitative indicators representing the number of axons capable of transmitting impulses and the relative extent of myelination in the axons, respectively [32], indicating that the reduced AMP and reduced CV could reflect polyneuropathy. Indeed, it has been reported that DPN-Check^®^ is a reliable and valid screening instrument to identify polyneuropathy [33]. However, little data were available on whether the reduction in CV or AMP measured by DPN-Check^®^ was more strongly associated with UAE in patients with type 2 diabetes. As shown in Figure 3, CV and AMP have a robust correlation with a Spearman’ Rho of 0.687; however, the results in Table 3 suggest that CV abnormalities may more accurately predict UAE, emphasizing the significance of detecting CV abnormalities early using DPN-Check^®^ in patients at increased risk for UAE.

We found that the association between DPN-Check^®^ and urinary albumin excretion is robust, particularly in women. Regarding diabetic neuropathy, there are no consistent research findings regarding whether men or women are more vulnerable to diabetic neuropathy [34,35,36,37,38]. Similarly, in our study, there was no difference in the prevalence of DPN-Check^®^-determined DPN between men and women (45 of 91 (50%) in women vs. 113 of 232 (49%) in men), nor in comorbidities, baseline laboratory parameters, physiological testing, or medications. Nonetheless, the reason why the association between DPN-Check^®^ and urinary albumin excretion is robust, particularly in women, is still completely unknown, and requires further study. Recently, a large-scale observational study conducted in Sweden revealed that women may be less likely than men to undergo monitoring of creatinine or albuminuria, or to seek adequate care for kidney disease [39]. Our results in this study may suggest that for these disadvantaged women, DPN-Check^®^ has the potential not only to diagnose diabetic neuropathy early, but also to identify patients at a high risk for kidney disease progression at an early stage.

Several limitations of our study should be acknowledged. First, the cross-sectional design of the study did not allow us to establish causality between DPN and UAE. Therefore, more longitudinal studies are needed in order to confirm whether patients with DPN are at a higher risk of a future decline in kidney function. Second, the sample size was relatively small, and as the data were obtained from East Asians, including Japanese individuals, it is not known whether the results of this study are applicable to other races. Third, CV and AMP measured by DPN-Check^®^ in this study have not been confirmed as accurate by NCS when performed by other methods. Therefore, although we were able to show that CV and AMP are associated with UAE when measured by the DPN-Check^®^, we were not able to confirm that they accurately diagnose DPN. Fourth, our study population comprised patients with type 2 diabetes, and the results may not be generalizable to patients with type 1 diabetes. Fifth, in our study, no association between DPN-Check^®^-determined DPN and eGFR could be observed in the multivariate analysis; however, the prevalence of patients classified as G3b was significantly higher in the group with DPN-Check^®^-determined DPN, presumably suggesting the possibility that DPN-Check^®^-determined DPN may be associated with lower eGFR. Further large-scale, long-term studies are needed in order to examine this association.

## 5. Conclusions

Our study found a significant association between diabetic peripheral neuropathy diagnosed using DPN-Check^®^ and urinary albumin excretion in patients with type 2 diabetes, particularly in female patients. However, since DPN-Check^®^ is a simplified electromyography device that observes the CV and AMP of the sural nerve, further research is needed in order to determine whether the abnormal action potential of the sural nerve is significantly associated with ACR, and whether abnormal CV and AMP of other peripheral nerves are significantly associated with ACR. Nevertheless, this finding suggests that DPN-Check^®^ could serve as a useful tool for identifying diabetic patients at risk for kidney damage, and may help to guide early interventions for both diabetic peripheral neuropathy and kidney complications. Further studies are needed in order to confirm our findings and to explore the potential clinical utility of DPN-Check^®^ in various settings.

## Figures and Tables

**Figure 1 jcm-12-04089-f001:**
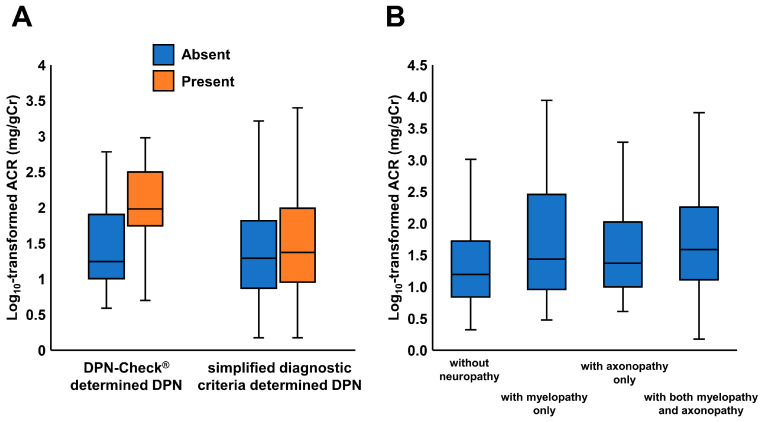
Urinary albumin-to-creatinine ratio (ACR) values in patients with and without DPN, as determined using each of the DPN-Check^®^ and simplified diagnostic criteria (**A**). ACR values in patients without neuropathy (n = 165), with myelopathy only (n = 25), with axonopathy only (n = 37), and with both myelopathy and axonopathy (n = 96) according to DPN-Check^®^ (**B**).

**Figure 2 jcm-12-04089-f002:**
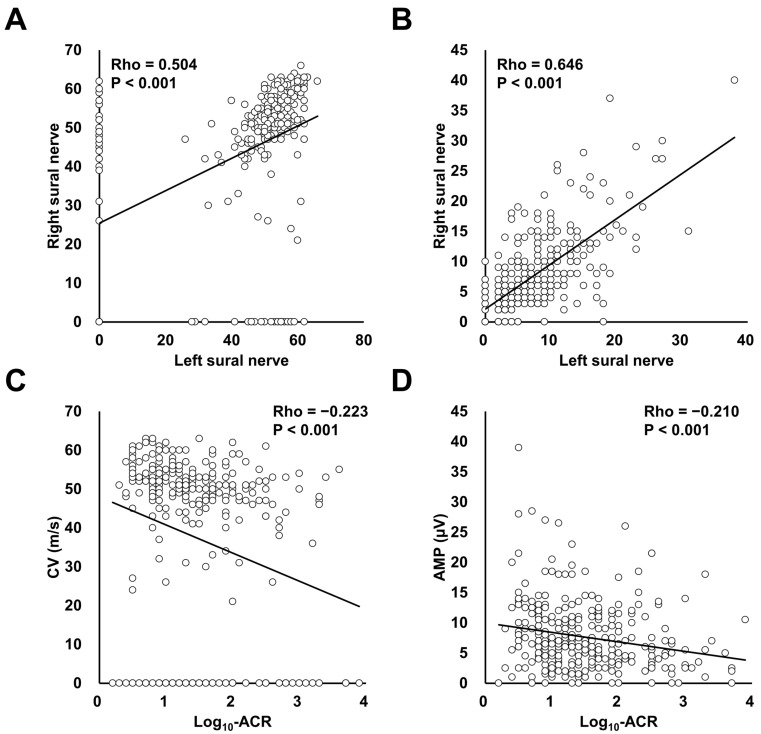
Spearman’s rank correlation coefficients of conduction velocity (CV, (**A**)) and amplitude (AMP, (**B**)) in DPN-Check^®^ for the left and right sural nerve, and of CV and AMP with log10-transformed albumin-to-creatinine ratio (Log-ACR), as shown in (**C**) and (**D**), respectively.

**Figure 3 jcm-12-04089-f003:**
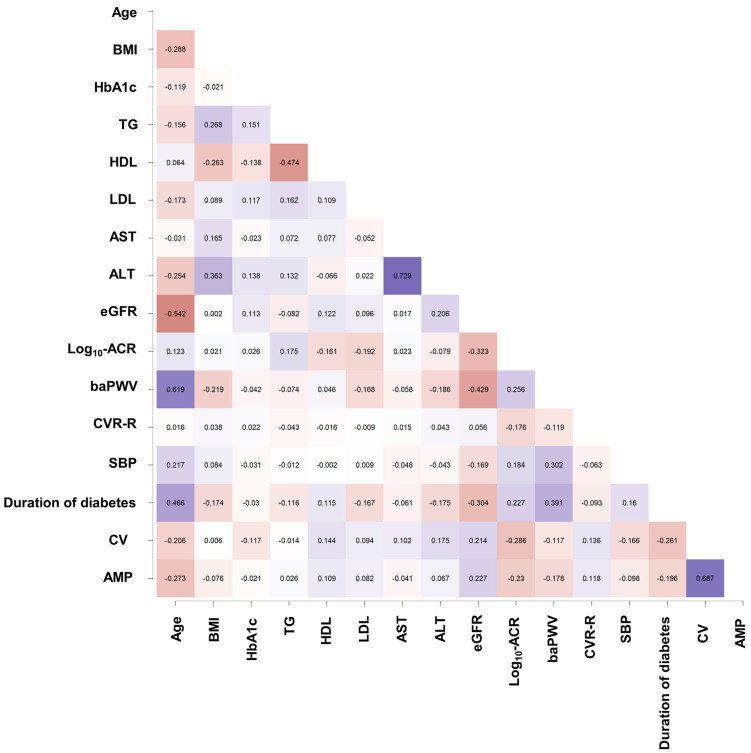
Spearman’s rank correlation coefficients of conduction velocity and amplitude in DPN-Check^®^, along with other clinical parameters. In the figure, negative values of Spearman’s correlation coefficient are shown in red and positive values in blue. Larger absolute values are indicated by darker colors.

**Table 1 jcm-12-04089-t001:** Baseline clinical characteristics, comorbidities, medication, laboratory parameters, and physiological testing in patients with and without DPN-Check^®^-determined DPN.

	Patients without DPN-Check^®^-Determined DPN	Patients with DPN-Check^®^-Determined DPN	*p* Value
	*n* = 165	*n* = 158	
Clinical characteristics at baseline			
Age, y	66.3 ± 11.9	66.0 ± 14.1	0.849
Male sex	119 (72)	113 (72)	0.904
Body mass index, kg/m^2^	24.3 ± 3.5	25.7 ± 4.9	0.002
Current smoker	29 (18)	37 (23)	0.904
Duration of diabetes	10.8 ± 8.9	14.6 ± 12.2	0.193
Comorbidities, n (%)			
CVD	19 (12)	44 (28)	<0.001
SDR	13 (8)	32 (20)	0.001
PDR	11 (7)	35 (22)	0.001
Baseline medications, n (%)			
Insulin	37 (22)	73 (46)	<0.001
Sulfonylureas	9 (5)	12 (8)	0.502
Metformins	106 (64)	84 (53)	0.043
Alpha-Gis	10 (6)	13 (8)	0.449
Glinides	18 (11)	16 (10)	0.858
TZDs	1 (1)	8 (5)	0.015
DPP4 inhibitors	66 (40)	59 (37)	0.624
SGLT2 inhibitors	80 (48)	78 (49)	0.874
GLP1-Ras	42 (25)	46 (29)	0.460
Imeglimin	0 (0)	5 (3)	0.027
ARBs	65 (39)	74 (47)	0.177
CCBs	51 (31)	55 (35)	0.455
Alpha blockers	1 (1)	2 (1)	0.537
Beta blockers	15 (9)	28 (18)	0.032
Diuretics	11 (7)	19 (12)	0.125
Statins	95 (58)	94 (59)	0.811
Fibrates	5 (3)	13 (8)	0.052
Ezetimib	22 (13)	29 (18)	0.226
EPAs	8 (5)	8 (5)	0.929
UA lowering agents	19 (12)	16 (10)	0.688
Anti-platelet agents	22 (13)	44 (28)	0.001
Laboratory parameters at admission			
HbA1c (%)	7.2 (6.7–8.1)	7.4 (6.8–8.6)	0.982
Triglycerides (mg/dL)	138 (89–194)	129 (93–210)	0.888
HDL cholesterol (mg/dL)	53 (43–65)	48 (41–60)	0.042
LDL cholesterol (mg/dL)	100 (78–120)	97 (74–115)	0.197
AST (IU/L)	20 (17–27)	20 (16–26)	0.761
ALT (IU/L)	20 (14–29)	18 (13–28)	0.255
Gamma-GTP (IU/L)	24 (14–45)	18 (13–28)	0.906
UA (mg/dL)	5.0 (4.4–6.0)	5.5 (4.5–6.4)	0.113
ACR (mg/g) (median)	15.7 (6.9–52.8)	34.9 (11.0–153.0)	<0.001
ACR (mg/g) (mean)	99.2 ± 390.9	339.2 ± 1034.3	0.006
Microalbuminuria (%)	47 (28)	52 (33)	0.384
Microalbuminuria (%)	11 (7)	32 (20)	<0.001
eGFR (mL/min/1.73 m^2^)	71 (60–80)	68 (50–80)	0.175
G1	13 (8)	15 (9)	0.616
G2	111 (67)	89 (56)	0.429
G3a	27 (16)	21 (13)	0.433
G3b	9 (5)	23 (15)	0.006
G4	5 (3)	10 (6)	0.159
Physiological testing			
SBP (mmHg)	128 ± 16	131 ± 22	0.142
DBP (mmHg)	73 ± 11	75 ± 15	0.161
CV (m/s)	54.5 (51.5–57.5)	29.5 (22.6–48.5)	<0.001
AMP (µV)	10.0 (7.0–13.0)	3.5 (2.0–5.3)	<0.001
CVR-R (%)	1.9 (1.4–3.0)	1.8 (1.1–2.7)	0.042
baPWV (cm/s)	1632 (1429–1932)	1628 (1427–1946)	0.948
Simplified diagnostic criteria-determined DPN	73 (44)	85 (54)	0.164
Symptom of DPN	62 (38)	95 (60)	0.259
Diminished Achilles tendon reflexes	86 (52)	72 (46)	0.163
Diminished vibratory sensation	84 (51)	74 (47)	<0.001

Data are expressed as mean ± standard deviation (SD), medians (interquartile range (IQR)), or numbers (proportion (%)). Comparisons of the groups in values are done using Chi-square test (percent), *t*-test (means), or Mann–Whitney U-test (medians). eGFR was calculated using the following formulas: Female with Cr ≤ 0.7 mg/dL, 144 × (Cr/0.7)^−0.329^ × 0.993^age^ × 0.813; female with Cr > 0.7 mg/dL, 144 × (Cr/0.7)^−1.209^ × 0.993^age^ × 0.813; male with Cr ≤ 0.9 mg/dL, 141 × (Cr/0.9)^−0.411^ × 0.993^age^ × 0.813; male with CR > 0.9 mg/dL; 141 × (Cr/0.9)^−1.209^ × 0.993^age^ × 0.813. GFR categories were defined according to KDIGO criteria as follows: G (grade) 1, eGFR ≥ 90; G2, eGFR 60–89; G3a, eGFR 45–59; G3b, eGFR 30–44; G4, eGFR 15–29 mL/min/1.73 m^2^. Abbreviations: ACR, urinary albumin-to-creatinine ratio; AMP, amplitude; ALT, alanine transaminase; ARBs, angiotensin receptor blockers; AST, aspartate transaminase; CCB, calcium channel blocker; CV, conduction velocity; CVD, cardiovascular disease; DBP, diastolic blood pressure; DPN, diabetic peripheral neuropathy; DPP4, dipeptidyl peptidase-4; eGFR, estimated glomerular filtration rate; EPA, eicosapentaenoic acid; Gis, glycosidase inhibitors; GLP1-RA, glucagon-like peptide-1 receptors agonist; GTP, glutamyl transpeptidase; HbA1c, hemoglobin A1c; HDL, high-density lipoprotein; LDL, low-density lipoprotein; PDR, proliferative diabetic retinopathy; SBP, systolic blood pressure; SDR, simple diabetic retinopathy; SGLT2, sodium–glucose cotransporter 2; TZDs, thiazolidinediones; UA, uric acid.

**Table 2 jcm-12-04089-t002:** The multiple regression analysis models examining the association between DPN-Check^®^-determined DPN and urinary albumin excretion.

	Total Patients (n = 323)		Male (n = 232)		Female (n = 91)	
	Standardized β	*p* Value	Standardized β	*p* Value	Standardized β	*p* Value
Univariate model	(Adjusted R^2^ = 0.055)		(Adjusted R^2^ = 0.046)		(Adjusted R^2^ = 0.089)	
DPN-Check^®^-determined DPN	0.235	<0.001	0.215	0.001	0.299	0.004
Age and Gender adjusted model	(Adjusted R^2^ = 0.063)		(Adjusted R^2^ = 0.047)		(Adjusted R^2^ = 0.026)	
DPN-Check^®^-determined DPN	0.236	<0.001	0.215	0.001	0.332	0.001
Age	0.078	0.149	0.004	0.780	0.281	0.005
Male sex	0.038	0.481	NA		NA	
Multivariate model	(Adjusted R^2^ = 0.338)		(Adjusted R^2^ = 0.347)		(Adjusted R^2^ = 0.438)	
DPN-Check^®^-determined DPN	0.123	0.012	0.085	0.145	0.214	0.010
eGFR	−0.319	<0.001	−0.332	<0.001	NA	
TG	0.209	<0.001	0.235	<0.001	NA	
PDR	0.127	0.010	0.136	0.020	NA	
SBP	0.155	0.001	0.157	0.005	NA	
Statins	0.100	0.033	NA	NA	NA	
GLP1-Ras	0.116	0.015	0.142	0.012	NA	
UA-lowering agents	NA		NA		0.269	0.001
Insulin use	0.143	0.004	0.125	0.034		
Beta blockers	NA		NA		0.166	0.053
CVR-R	NA		NA		−0.141	0.049
Duration of diabetes	NA		NA		0.380	<0.001

Abbreviations: CVR-R, coefficient of variation of RR intervals; DPN, diabetic peripheral neuropathy; eGFR, estimated glomerular filtration rate; GLP1-RA, glucagon-like peptide-1 receptor agonist; NA, not applicable; PDR, proliferative diabetic retinopathy; SBP, systolic blood pressure; TG, triglyceride; UA, uric acid.

**Table 3 jcm-12-04089-t003:** The multivariate regression analysis models examining the association between abnormalities in conduction velocity (CV), amplitude (AMP), and albumin-to-creatinine ratio, respectively.

	Total Patients (n = 323)	
	Standardized β	*p* Value
Univariate model	(Adjusted R^2^ = 0.056)	
Abnormal CV	0.162	0.018
Abnormal AMP	0.101	0.139
Age- and Gender-adjusted model	(Adjusted R^2^ = 0.063)	
Abnormal CV	0.150	0.029
Abnormal AMP	0.113	0.100
Age	0.069	0.403
Male sex	0.046	0.403
Multivariate model	(Adjusted R^2^ = 0.346)	
Abnormal CV	0.128	0.022
Abnormal AMP	0.023	0.670
eGFR	−0.310	<0.001
TG	0.202	<0.001
PDR	0.129	0.008
SBP	0.154	<0.001
Statins	0.095	0.044
GLP1-Ras	0.110	0.021
Insulin use	0.137	0.006

Abbreviations: AMP, amplitude; CV, conduction velocity, eGFR, estimated glomerular filtration rate; GLP1-RA, glucagon-like peptide-1 receptors agonist; PDR, proliferative diabetic retinopathy; SBP, systolic blood pressure; TG, triglyceride.

## Data Availability

The data underlying this article will be shared upon reasonable request to the corresponding author.

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
