# Peer review of "Association between Diabetic Peripheral Neuropathy as Measured Using a Point-of-Care Sural Nerve Conduction Device and Urinary Albumin Excretion in Patients with Type 2 Diabetes"

_jcm, 2023, doi:10.3390/jcm12124089_

Round 1

Reviewer 1 Report

There is no doubt that neuropathy, retinopathy and nephropathy develop in many type 2 diabetic patients as a result of chronic hyperglycemia. Although in many cases, neuropathy and retinopathy can develop long before the onset of diabetes. In terms of frequency, neuropathy ranks first, retinopathy second, and nephropathy third.

Many studies have shown that electromyographic changes in the peripheral nerves of the lower extremities correlate with the degree of compensation for diabetes mellitus. From the peripheral nerves n. suralis is the first to respond to decompensation of diabetes mellitus. Diabetic neuropathy can have an axonopathic character, a myelinopathic character, or an axnomyelinopathic character.

Thus, in the initial stages of the disease, a correlation with decompensatory reactions can be identified with one of the electromyographic indicators. For example, with the amplitude of the action potential in axonopathic neuropathy or with the conduction velocity in myelinopathic neuropathy. At the same time, the conduction velocity  does not correlate with decompensatory  reactions in axonopathic neuropathy and the amplitude of the action potential does not correlate with decompensatory  reactions in patients with myelinopathic neuropathy. In axonomyelinopathic neuropathy, a correlation can be determined with the amplitude of the action potential and with conduction velocity.

Your work is very interesting and is of significant scientific and practical interest. Moreover, the use of such a device increases the possibility of early neurophysiological diagnosis of diabetic neuropathy.

The relevance of this problem is determined by the large number of patients with type 2 diabetes mellitus, severe complications and the rarity of the use of electromyography in many countries and cities, and the lack of trained specialists for this purpose.

Question 1: Was there an age limit in your work? Indicated from 20 years old, until which is not indicated.

Question 2: Was there a distribution of patients according to the nature of EMG disorders: axonopathic, myelinopathic, axonomyelinopathic before conducting a correlation analysis with the amplitude of the action potential or with the conduction velocity.?

Question 3: was correlation analysis performed with pathological changes in the amplitude of the action potential and conduction velocity or with all the data obtained, even with the norm? It is important to note that under normal conditions, the rate and amplitude of the action potential can be very variable.

Question 4: You indicated in this study results that you found a significant association between DPN-Check® diabetic peripheral neuropathy and urinary albumin excretion in patients with type 2 diabetes. Here it is necessary to indicate that these are electromyograpic changes detected using the DPN-Check®. Because the DPN-Check® is a simplified electromyography device, the changes found in n. Suralis are neurophysiological or electromyograpic.

Question 5. In the introduction you noted that the prevalence of diabetic neuropathy is 50%. Depending on how much EMG data you have correlated in your study, you should report the prevalence of diabetic neuropathy detected by EMG. It is also necessary to indicate the prevalence of albuminuria for comparison.

Question 6: in lines 61-62: there are few reports examining the association between DPN diagnosed by the DPN-Check® and urinary albumin excretion. Please post links.

Question 7: It is necessary to indicate in the introduction or in the discussion about the conducted studies proving that the data obtained using the DPN-Check® do not differ from the results of standard electromyography. This is important for the credibility of the study. Since you are examining the amplitude of the action potential and the conduction velocity of the n.suralis impulses, there is no new information obtained using the DPN-Check device.

Question 8: In lines 118-120. by one nurse (A.F.) specialized in diabetes care under the direction and supervision of diabetologists certified by JDS. It is necessary to clarify if diabetologists have completed a course in electromyography.

Question 9:  In lines 121-122: Un- detectable nerve conduction was considered AP 0 μV and CV 0 m/s for DPN diagnosis. Undetectable nerve conduction due to the lack of an action potential response, this means a decrease in the amplitude of the action potential and does not mean that the conduction velocity = 0. But, given the results indicated in Table No. 1, the conduction velocity is  (22.6–48.5) m/ s and an action potential amplitude range (2.0–5.3) mcV , that is, there were no such patients with undetectable results.

Question 10: Can you indicate, using mean averages, how many cases of albuminuria were detected among patients with neuropathy and among patients without neuropathy. It would also be interesting if you compared the severity of albuminuria in patients with severe neuropathy with the severity of albuminuria in patients with moderate neuropathy. It would also be interesting if these changes were compared in patients with a decrease in conduction velocity and in patients with a decrease in the amplitude of the action potential.

Question 11: The references are very old, and 13 of them are older than 2006. Can you update the sources for more recent ones?

Reviewer 2 Report

Studies have shown that diabetic neurosis is associated with albumin excreta in the urine of patients with type 2 diabetes. In particular, these studies used a device called DPN ® to detect peripheral blood diabetic neuropathy. The study was retrospective, which is a limitation that does not allow a causal relationship to be established between markers of neuropathy and nephropathy. The methodology was well described and the conclusions matched the results, but when the article was published, a number of comments appeared:

1. It must be pointed out, however, that despite the retrospective nature of the study and the impossibility of signing an IS specifically for this protocol, patients nonetheless signed an informed consent for treatment and processing of personal information when applying to the hospital. If this is not the case, then the study has no legal basis. And if there is such an IS, then it is incorrect to write about the absence of any IS.

2. The list of literature that is analyzed is mainly older than 10 years (73.5%), only 2 sources of the last 5 years (5%) and 26.5% of the sources were published during the last 10 years. I think that this makes the discussion not relevant enough.

3. Why was the formula proposed by the Japanese Society of Nephrology chosen to calculate glomerular filtration rate, and not CKD-EPI (Chronic Kidney Desease Epidemiology Collaboration), which has been recommended for KDIGO filtration assessment since 2012 and has also been tested on the Japanese population? Has GFR been associated with measures of neuropathy?

4. An important note is the fact that there is no clinical assessment of patients is very limited, in particular, and in the article on diabetic kidney disease, there is no indication of the existence of a generally accepted classification of chronic kidney disease and the stratification of patients on this basis.

5. I hope the authors will address the question in the discussion as to why the association between dpn and akr is weakened by men in the gender analysis. The authors think this has something to do with the methods used to diagnose diabetic peripheral neuropathy, specifically using a method for treating psychosis? Or what? Unfortunately, this fact has been completely ignored in the discussion.

6. Is the association between diabetic neuropathy and complications such as kidney disease new? As far as I know, this link has been studied.

Round 2

Reviewer 1 Report

The work is of great practical and scientific interest. The topic is very relevant, especially considering the annually growing number of patients with diabetes mellitus and the growing number of complications with it, including diabetic neuropathy.

For early diagnosis of diabetic neuropathy, regular examination of the peripheral nerves of the lower extremities using electromyography is recommended.

However, there are not so many specialists who could conduct electromyography for a large number of patients with diabetes mellitus. In this regard, the implementation in clinical practice of such a device for the study of n.suralis as DPN-Chek  device is a successful solution to this problem and opens up great opportunities for the prevention of complications of diabetes mellitus and for the study of patients with diabetic neuropathies.

For the first time such an analysis was carried out between the severity and characteristics of neuropathy on the one hand and the severity of albuminuria on the other hand.

The purpose of the study is described clearly.

The rational for selecting that particular statistic, and which variables were entered into the statistic are described.

Statistical results are presented in a Table or Figure.

The introduction discusses the main issues related to the topic, with reference to contemporary sources. The results are presented and demonstrated in an understandable scientific language using comparative and correlation analysis. Results are structured around the Research Questions.

Based on the results of this study, the authors came up with a method for determining the degree of compensation of diabetes mellitus and early diagnosis of complications of this disease.

The title of the article matches the content. The purpose and objectives of the work are fully realized.

Discussion and conclusions follow logically from the results of the study and are fully consistent with the purpose of the study.

The main findings as related to the overall purpose of the study are discussed and explained in detail.

Conclusions is directly related to the data that was collected and analyzed.

Author Response

We greatly appreciate your meaningful comments. It has improved the quality of our paper.

Reviewer 2 Report

The authors did a great job and improved the article according to all the comments. I was completely satisfied with the answers of the authors and the work done.

I only propose to add to the notes to Table 1 the decoding of the abbreviations G1-G5, and the indication of the eGFR formula, despite the fact that this has already been added to the materials and methods.

Author Response

I would like to thank you for your comments that have brought about appropriate and meaningful improvements. 

According to the comment of "I only propose to add to the notes to Table 1 the decoding of the abbreviations G1-G5, and the indication of the eGFR formula, despite the fact that this has already been added to the materials and methods.", we added eGFR formula and Abbreviation of G1-G5 in Line 193.